# Circulating miR-499a and miR-125b as Potential Predictors of Left Ventricular Ejection Fraction Improvement after Cardiac Resynchronization Therapy

**DOI:** 10.3390/cells11020271

**Published:** 2022-01-13

**Authors:** Isabel Moscoso, María Cebro-Márquez, Álvaro Martínez-Gómez, Charigan Abou-Jokh, María Amparo Martínez-Monzonís, José Luis Martínez-Sande, Laila González-Melchor, Javier García-Seara, Xesús Alberte Fernández-López, Sandra Moraña-Fernández, José R. González-Juanatey, Moisés Rodríguez-Mañero, Ricardo Lage

**Affiliations:** 1Cardiology Group, Centre for Research in Molecular Medicine and Chronic Diseases (CIMUS), Universidade de Santiago de Compostela, 15782 Santiago de Compostela, Spain; isabel.moscoso@usc.es (I.M.); Maria.Cebro.Marquez@sergas.es (M.C.-M.); Jose.Ramon.Gonzalez.Juanatey@sergas.es (J.R.G.-J.); 2Department of Cardiology and Coronary Unit and Cellular and Molecular Cardiology Research Unit, Institute of Biomedical Research (IDIS-SERGAS), University Clinical Hospital, 15706 Santiago de Compostela, Spain; Alvaro.Martinez.Gomez@sergas.es (Á.M.-G.); Charigan.Abou.Jokh.Casas@sergas.es (C.A.-J.); Maria.Amparo.Martinez.Monzonis@sergas.es (M.A.M.-M.); Jose.Luis.Martinez.Sande@sergas.es (J.L.M.-S.); Laila.Gonzalez.Melchor@sergas.es (L.G.-M.); Javier.Garcia.Seara@sergas.es (J.G.-S.); jesus.alberto.fernandez.lopez@sergas.es (X.A.F.-L.); sandra.morana.fernandez@usc.es (S.M.-F.); Moises.Rodriguez.Manero@sergas.es (M.R.-M.); 3Centro de Investigación Biomédica en Red de Enfermedades Cardiovasculares (CIBERCV), 28029 Madrid, Spain

**Keywords:** heart failure, adverse remodelling, cardiac resynchronization therapy, microRNAs, cardiac biomarker, cardiac therapeutic target

## Abstract

Cardiac resynchronization therapy represents a therapeutic option for heart failure drug-refractory patients. However, due to the lack of success in 30% of the cases, there is a demand for an in-depth analysis of individual heterogeneity. In this study, we aimed to evaluate the prognostic value of circulating miRNA differences. Responder patients were defined by a composite endpoint of the presence of left ventricular reverse remodelling (a reduction ≥15% in telesystolic volume and an increment ≥10% in left ventricular ejection fraction). Circulating miRNAs signature was analysed at the time of the procedure and at a 6-month follow-up. An expression analysis showed, both at baseline and at follow-up, differences between responders and non-responders. Responders presented lower baseline expressions of miR-499, and at follow-up, downregulation of miR-125b-5p, both associated with a significant improvement in left ventricular ejection fraction. The miRNA profile differences showed a marked sensitivity to distinguish between responders and non-responders. Our data suggest that miRNA differences might contribute to prognostic stratification of patients undergoing cardiac resynchronization therapy and suggest that preimplant cardiac context as well as remodelling response are key to therapeutic success.

## 1. Introduction

Heart failure (HF) causes thousands of deaths per year worldwide. Irrespective of the underlying cause, HF is a shared chronic phase in the progression of many cardiovascular diseases (CVD). Untreated mild dysfunction progresses to severe HF as a result of progressive structural remodelling, characterized by cardiac hypertrophy, alteration of extracellular matrix homeostasis, fibrosis, metabolic abnormalities, defective autophagy, mitochondrial dysfunction, apoptosis, and left ventricular ejection fraction (LVEF) decrease [1]. Despite significant advances in medical treatment, many HF patients remain refractory to conventional drug therapy. When dyssynchrony is suspected in HF patients, cardiac resynchronization therapy (CRT) emerges as a therapeutic approach to restore cardiac synchronization. This can be accomplished using biventricular pacing, i.e., a pacing lead in the right ventricle and a second lead in the lateral wall of the left ventricular (through an epicardial vein). By re-establishing cardiac synchrony, hemodynamics may be improved due to the optimization of the interventricular and intraventricular contraction sequences, thereby, potentially decreasing mitral regurgitation, reducing pulmonary venous pressure, increasing myocardial preload, and improving cardiac output [2,3]. CRT has gained much acceptance as an additional therapeutic option for patients with severe dysfunction (LVEF ≤35%), electrical dyssynchrony (QRS duration ≥150 ms), and persistent HF symptoms. Unfortunately, the mechanisms underlying the CRT response are poorly understood and around 30% of patients are non-responders [4]. Non-adaptive remodelling is key in a poor prognosis. Despite current accumulated knowledge, we lack prognostic markers that allow an adequate risk stratification and an improved success rate [5].

MicroRNAs (miRNAs) are small single-stranded ribonucleotides, involved in post-transcriptional regulation of gene expression [6]. Under pathophysiological conditions, it is believed that cells release some of their miRNA content into the circulation [7,8,9,10]. Recently, they have been widely studied in CVD [11], suggesting that miRNA signature might be more representative than classic cardiac biomarkers [9]. Selective miRNA regulation has been associated with pathophysiogical processes such as altered adrenergic response [12], contractile and electrophysiological abnormalities [13,14], fibrosis [15], and cardiac hypertrophy [16]. In HF patients, miRNAs suggest a key role modulating cardiac metabolism, ventricular dysfunction [17], fibrosis, and cardiac remodelling [18,19]. Transcardiac gradient of miRNA in HF patients, due to the selective myocardial secretion and absorption, emphasizes the pathophysiological role in CVD [20].

Among dysregulated miRNAs in CVD, miR-499 and miR-125b-5p have been closely involved in HF; both have been described as highly expressed cardiac miRNAs, which are released under stressful conditions [10,21,22] as well as good circulating biomarkers in HF [22,23].

The current data suggest a prognostic value of miRNA signature in CRT patients. Altered miRNA profiles correlate with cTnI levels [24] and are involved in cardiac hypertrophy, fibrosis, and apoptosis [25]. In addition, recent data suggest that preimplant miRNAs profiles might predict recovery potential in non-ischemic cardiomyopathy patients [26]. In this study, we aim to evaluate whether: (a) baseline miRNA profile differences might predict CRT response, (b) follow-up changes are associated with functional improvement, and (c) altered miRNAs might be involved in the CRT response.

## 2. Materials and Methods

### 2.1. Study Design and Subjects

Symptomatic HF patients who were referred for CRT implantation were consecutively recruited from the Cardiovascular Unit at the University Hospital of Santiago de Compostela. The inclusion criteria were as follows: age older than 18 years, NYHA class II to IV despite optimal medical treatment for 6 months, LVEF <35%, and QRS duration >150 ms. A left bundle branch block was defined according to Strauss´s criteria [27]. A 12-lead ECG, systolic and diastolic blood pressure, body weight, two-dimensional and Doppler echocardiography, and NYHA functional class were routinely assessed. The exclusion criteria included: a previous history of suboptimal HF medical therapy, acute myocardial infarction (AMI), planned revascularization, and expected survival <6 months. Patients with a known history of leukopenia, thrombocytopenia, or severe hepatic or renal dysfunction, as well as evidence of inflammatory or malignant disease were also excluded. In order to accurately assess the changes associated with an effective response to CRT, patients with suboptimal lead position (lateral/posterolateral vein of the basal-medium region of LV) were excluded. However, a high LV threshold was not considered to be an exclusion criterion as long as the LV lead could be placed in an “optimal” location. All patients were evaluated at baseline and 6 months after device implantation. To avoid selection bias, responder patients were classified by a composite endpoint of the presence of left ventricular reverse remodelling (a reduction ≥15% in telesystolic volume and an increment ≥10% in left ventricular ejection fraction) but not NYHA changes. Patients were grouped according to CRT response, prior to miRNA profile determination. The study complies with the Declaration of Helsinki and was approved by the Clinical Research Ethics Committee of Galicia (MRM-miRHF-2017-01). All patients signed an informed consent.

### 2.2. Echocardiographic Evaluation

The electro- and echocardiographic parameters included QRS width and morphology, LV end-diastolic (LVEDV) and end-systolic volume (LVESV), LVEF, and left atrial diameter (LAD). CRT success was defined by improvements in telesystolic volume (a reduction ≥15%) and LVEF (an increment ≥10%). All echocardiographic examinations were carried out by using the General Electric (GE) Vivid 7 Dimension (General Electric Company, Horten, Norway) and performed by the same operator to avoid interobserver error. Patients were classified as responders based on CRT success, independent of NT-proBNP or NYHA class improvements.

### 2.3. Device Implantation

The device was implanted in an electrophysiology operating room. All patients underwent an angiography of the coronary venous system, and the lead was implanted in the lateral/posterolateral vein of the basal-medium region of the LV. Proper biventricular stimulation was systematically verified at discharge, at 3 and 6 months. All studies were performed in the morning, and medications were maintained according to current recommendations [28].

### 2.4. Blood Collection

Blood samples were obtained in the morning of the CRT procedure, before device implantation, and 6 months after implantation using standard methods. Samples were immediately processed and frozen at −80 °C for 6 to 18 months until use.

### 2.5. RNA Extraction and miRNA Quantification

The total RNA was extracted using an miRNeasy Serum/Plasma Advanced Kit (Qiagen, Hilden, Germany). Briefly, cDNA was transcribed from the extracted RNA using an miScript II RT kit (Qiagen, Hilden, Germany) in a SimpliAmp Thermal Cycler (Applied Biosystems, Carlsbad, CA, USA). The cDNA was preamplificated using an miScript PreAMP PCR Kit (Qiagen, Hilden, Germany) with a miScript PreAmp Universal Primer and Human Cardiovascular Disease miScript PreAmp Pathway Primer Mix (MBHS-113Z, Qiagen, Hilden, Germany), and then, to perform miRNA quantification by polymerase chain reaction (RT-qPCR), was aliquoted into the Human Cardiovascular Disease miScript miRNA PCR Array containing 84 different predesigned mature miRNAs (MIHS-113Z, Qiagen, Hilden, Germany) with an miScript SYBR Green PCR kit (Qiagen, Hilden, Germany). The miRNAs are listed in Appendix A. All cDNA steps and PCR setup were performed by a QuantStudio™ 7 Flex Real-Time PCR System, 384-well (Applied-Biosystems, Carlsbad, CA, USA). The PCR cycling was performed according to the manufacturer’s protocol and conditions. Individual miRNAs were determined to be detected when Ct values were lower than 30; Ct value ≥30 were considered not detected.

Normalized miRNA expression at baseline and 6 months post-CRT are represented by ΔCt, that were calculated by subtracting the global geometric mean signal of miRNAs that were commonly expressed in the present study from individual miRNA Ct values. Commonly expressed miRNAs were identified as those with Ct values <30 in all samples, utilizing the miScript miRNA PCR Array Data Analysis Tool (Qiagen, Hilden, Germany). Fold change was calculated as 2^−∆∆Ct^.

### 2.6. MicroRNA Pathway Analysis and Target Prediction

The GO terms and KEGG pathway annotation analyses were performed using an miRNet analysis tool (https://www.mirnet.ca, accessed on 19 October 2021), which utilized data from well-annotated databases: miRTarBase v8.0, TarBase V8.0, and miRecords. The analyses were performed to fully understand the functional role of differentially regulated miRNAs.

### 2.7. Statistical Analysis

Paired and unpaired *t*-tests were used to detect the differences in miRNA expression between CRT groups using the Qiagen array-analysis tool (Qiagen, Hilden, Germany). The Shapiro-Wilk test was performed to test normality of distribution. The Pearson correlation test, AUC ROC analysis, and ANOVA followed by Tukey’s post hoc test were performed with GraphPad Prism 9 (GraphPad Software Inc., San Diego, CA, USA). In all analyses, a two-tailed *p* < 0.05 was considered to be significant.

## 3. Results

### 3.1. Study Population

Twenty-eight patients who were undergoing CRT were consecutively included. Patients with suboptimal lead implantation were excluded. Twenty-five patients were included and classified as responders (56%) or non-responders (44%) (Figure 1).

Demographic characteristics are described in Table 1. No differences were found in preimplant clinical, electrocardiographic, and echocardiographic determinant statuses among both groups (Table 1). At follow-up, responders showed a significant improvement in, NYHA (1.64 ± 0.50 vs. 2.43 ± 0.51, *p* = 0.007), LVEF (46 ± 12% vs. 30 ± 8, *p* = 0.002), LVEDV (102 ± 44 vs. 187 ± 101 mL/m^2^, *p* = 0.003), LVESV (59 ± 38 vs. 138 ± 87 mL/m^2^, *p* = 0.001), and NT-proBNP levels (1744 ± 1481 vs. 3361 ± 2697 pg/mL, *p* = 0.002). In contrast, non-responders showed a significant increase in NT-proBNP levels (5254 ± 5325 vs. 12914 ± 29,691 pg/mL, *p* = 0.001). A significant reduction in the QRS interval was observed in both, responders (172 ± 24 vs. 136 ± 23 ms, *p* = 0.008) and non-responders (164 ± 32 vs. 142 ± 27 ms, *p* = 0.024) (Table 1).

### 3.2. Baseline Differences and Longitudinal Changes in miRNA Profiles after CRT

The baseline assessment of miRNA profiles showed a significant decrease in miR-107 (0.47-fold decrease, *p* = 0.037) and miR-499a-5p (0.5-fold decrease, *p* = 0.033) of responders (Figure 2 and Appendix A).

### 3.3. Follow-Up Changes in miRNA Profiles after CRT

Individual analysis of miRNA expression relative to baseline profile showed a significant decrease in circulating levels of miR-125b-5p (0.34-fold decrease, *p* = 0.031) and miR-378a-3p (0.55-fold decrease, *p* = 0.002) in responders (Figure 3 and Appendix A). No statistical differences were found in the 6-month responder vs. non-responder analysis (Appendix A) or in the longitudinal analysis of non-responder patients (Appendix A).

### 3.4. MicroRNAs Predictive Value of CRT Response

A ROC analysis was performed to assess the predictive value of significative regulated miRNAs as biomarkers of CRT response (Figure 4 and Appendix A). MiR-107 (AUC of 0.740 [0.543 to 0.937], *p* = 0.043) and miR-499a-5p (AUC of 0.747 [0.539 to 0.953], *p* = 0.037), regulated at baseline as well as miR-125b-5p (AUC of 0.735 [0.546 to 0.924], *p* = 0.035) and miR-378a-3p (AUC of 0.816 [0.654 to 0.979], *p* = 0.004), regulated at follow-up, showed a marked sensitivity and specificity for CRT success. (Figure 4a,b and Appendix A).

### 3.5. MicroRNA Correlation between Echocardiographic and Clinical Parameters

A correlation analysis of echocardiographic data and significative baseline and follow-up regulated miRNAs was performed. LVEF improvement negatively correlated with the baseline expression on miR-499-5p (*r* = −0.420, *p* = 0.039) (Figure 5a) but also with the observed relative decrease in miR-125b-5p expression at follow-up (*r* = −0.454, *p* = 0.029) (Figure 5b). Neither correlation was found with miR-107 and miR-378a-3p (data not shown). No significant associations were found between regulated miRNAs and other echocardiographic or clinical parameters.

### 3.6. Predictive Analysis of miR-499-5p and miR-125b-5p in CRT Patients

Among others, regulated miR-499-5p and miR-125b-5p were predicted to target multiple genes involved in several pathways related to cardiovascular disease. The KEGG analysis included over-representation of pathways related to apoptosis, TGF-β, mTOR, insulin, MAPK, p53 signalling pathways, as well as Wnt and adipocytokine signalling pathways (Figure 6).

## 4. Discussion

In the present study, we aimed to characterize circulating miRNA profile involved in functional response to CRT. Our results showed that LVEF improvement correlated with lower baseline levels of miR-499 and a selective decrease in miR-125b-5p at follow-up. Our data showed preimplant and follow-up differences in miRNA profiles associated with functional improvement after CRT that might contribute to prognostic stratification of HF patients. These data reinforce the prognostic role of plasma miRNAs in CRT and also suggest that they might play a functional role in the pathogenesis of HF.

To date, most miRNA analyses have compared HF patients against healthy subjects [25,29]; we aimed to characterize the prognostic value of baseline differences as well as individual changes over time in recommended CRT patients.

Although CRT is a well-validated approach in patients with HF, our cohort exhibited a high rate of non-responders, even higher than previous data [4]. Our rate seems to be associated with a strict echocardiographic definition of response (with a reduction in telesystolic volume ≥15% but an increment in LVEF >10%) and the exclusion of symptomatic change as criteria for response, in order to accurately assess the changes associated with an effective response to CRT.

MicroRNAs have been linked to pathophysiology responses in cardiovascular disease [11], including structural and functional changes in a failing heart [19,26,30]. As compared with previous data showing a positive correlation between baseline plasma miR-30d level and CRT response in heart failure patients with dyssynchrony [31], our data revealed that only basal levels of miR-499a-5p correlated with LVEF improvement.

MiR-499 is a muscle-specific miRNA [32], mainly expressed in the myocardium [21], involved in myoblast proliferation and differentiation [33]. Increased levels of miR-499 have been described in patients and preclinical models of AMI [34,35,36] and patients with acute HF decompensation [10] but not with chronic HF. Elevated miR-499-5p plasma levels, following an MI, negatively correlate with systolic function [21]. Recent data have shown that miR-499 is specifically released from the heart [21] under stressful conditions [10].

Increased levels of miR-499 have been associated with prolonged aerobic exercise in healthy runners, returning to baseline levels after 24 hours at rest [37,38]. Coherently, miR-499 levels have been negatively correlated with VO2 peak in HF subjects [39]. Recent data demonstrate a dose-dependent relationship between miR-499 expression and cardiac stress response, suggesting that a tight control of miR-499 levels is key in cardiovascular pathophysiology [40]. All these data suggest that circulating levels of miR-499 might serve as an indicator of myocardial status in HF [39]. The AUC analysis showed high sensitivity and predictive capacity of miR-499 differences between CRT responders and non-responders. Our data also showed that miR-499-5p expression correlated with LVEF improvement. This correlation was previously reported in AMI patients, where the authors found that plasma levels of circulating miR-499 could contribute to the identification of a high risk of patient death after AMI [21,41]. Coherently, lower baseline plasma levels of miR-499 might reflect a better myocardial condition, acting as a prognostic factor. No other clinical biomarker correlates with miR-499-5p expression (baseline or follow-up).

In addition to baseline differences, our results showed a significant decrease in miR-125b-5p at follow-up that correlated with LVEF improvement. No significant associations were found between miR-125b-5p and other clinical biomarkers. Higher circulating levels of miR-125b have been described in HF patients as compared with no-HF patients [22,29,42]. Interestingly, recent data suggest that miR-125b might be secreted by cardiac tissue [22]. In this sense, miR-125b has been identified as one of the five most abundant miRs in pericardial fluid of HF patients [43] and coronary sinus blood levels of miR-125b double peripheral levels in HF patients [22]. Previous data suggested that miR-125b may cause HF by early hypertrophy and cardiac remodelling [44]. MiR-125 has been shown to modulate fibrotic response by repressing several antifibrotic mechanisms [30,45,46]. Myocardial fibrosis has been identified as a prognostic factor in patients with non-ischemic HF [47] and a predictor of response to CRT [48], suggesting that low fibrosis biomarkers might act as CRT response predictors. Our results showed a marked decrease in miR-125 levels at follow-up. The AUC value also showed a marked sensitivity of miR-125b decrease as a predictor of HF improvement. Coherently, functional recovery is associated with a significant decrease in miR-125b. These data, according to reverse remodelling, suggest that an adequate profibrotic response might determine resynchronization therapy success.

In silico analyses have predicted that several overrepresented pathways, regulated by differentially expressed miR-499-5p and miR-125b-5p, are involved in the pathophysiology of HF. The TGF-β, Jak-STAT, and Wnt signalling pathways play a key role in cardiac fibrosis and cardiac remodelling [49,50,51], while MAPK and neurotrophin signalling are associated with cardiomyocyte growth and cardiac hypertrophy [52,53]. Similarly, increased apoptosis [54,55], altered insulin [56] and adipocytokine signalling have been well established in patients with heart failure [57,58].

The present study has some limitations. First, the study was conducted in a single centre with a small sample size. Since the study was performed to assess the early performance of biomarkers, the follow-up period was extended until the moment at which patients could be classified by CRT response. Future studies should be extended up to 12 months to assess associations among circulating miRNA changes and LVEF as well as long-term predictive capacity. Although all the echocardiographic studies were always performed by the same experienced operator, accredited by the European Society of Cardiology in Echocardiography, and always with the same device, intraobserver error must always be considered in data interpretation. In addition, although the patients were well characterized, the clinical value of miRNAs as biomarkers for CRT success should be compared with classical predictive parameters (sex, CVD, bundle branch block type, underlying rhythm, etc.), which would require a larger sample size. Our exploratory data considered the myocardial substrate characteristics that could be related to CRT response; however, for translation into clinical practice it would be necessary to propose a clinical trial, including a larger sample size and healthy controls that could confirm our hypothesis. Our results also suggest that selective miRNA regulation by using anti-miRNAs, miRNAs, or miR-mimetic sponges might constitute a therapeutic approach. Whether these pathways truly constitute a therapeutic approach needs to be determined by laboratory experiments and is open for future studies.

## 5. Conclusions

In conclusion, our data demonstrates that baseline miR-499a and also follow-up miR-125b levels correlate with LVEF improvement after cardiac resynchronization therapy, supporting the use of miRNAs as predictive biomarkers for CRT response. In addition, selective miRNA differences suggest that CRT success might depend on basal cardiac performance as well as an adequate myocardial anti-fibrotic response (Figure 7).

## Figures and Tables

**Figure 1 cells-11-00271-f001:**
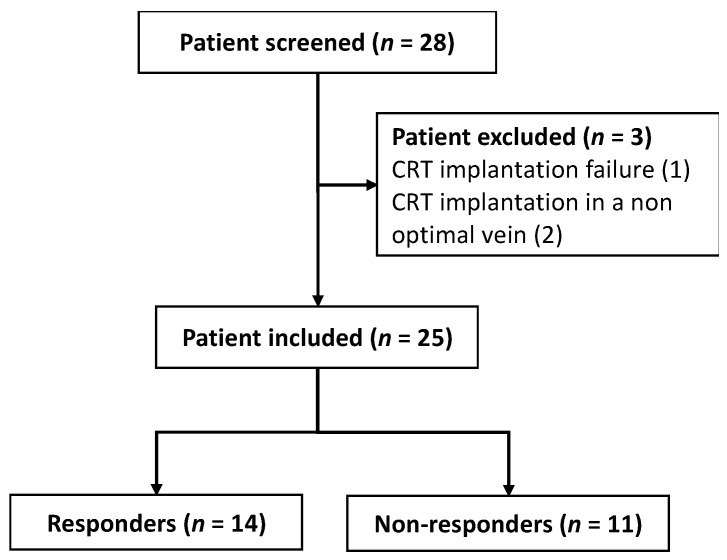
Study flow chart.

**Figure 2 cells-11-00271-f002:**
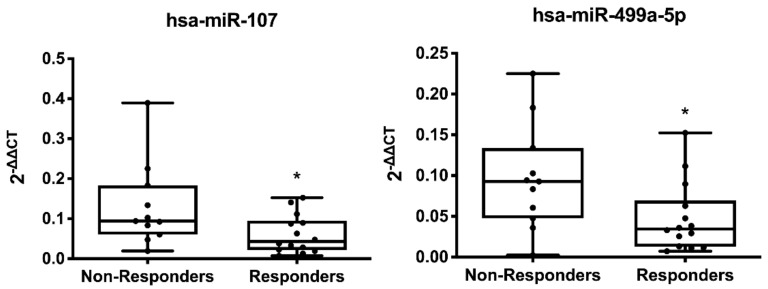
Differential baseline expression of miRNAs between CRT responders and non-responders. Box-and-whiskers min to max plots showing plasma levels of microRNAs. Data correspond to baseline differentially expressed miRNAs in non-responder (*n* = 11) vs. responder (*n* = 14) patients. * *p* < 0.05 responders vs. non-responders.

**Figure 3 cells-11-00271-f003:**
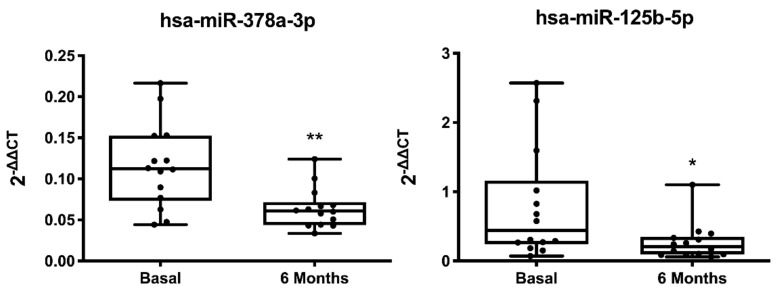
Differential follow-up expression of miRNAs in responders. Box-and-whiskers min to max plots showing plasma levels of microRNAs. Data correspond to differentially expressed miRNAs in responder patients (*n* = 14). * *p* < 0.05 and ** *p* < 0.01 basal vs. 6 months.

**Figure 4 cells-11-00271-f004:**
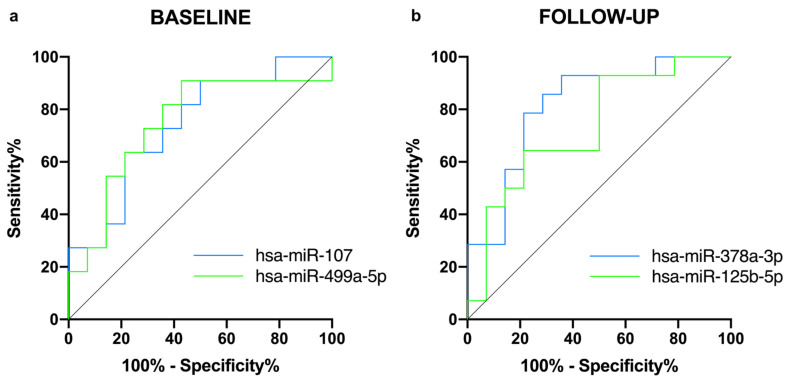
Predictive capacity of CRT success. Receiver-operating characteristic (ROC) curves comparing sensitivity and specificity of (**a**) baseline and (**b**) follow-up differentially expressed miRNAs in predicting CRT response.

**Figure 5 cells-11-00271-f005:**
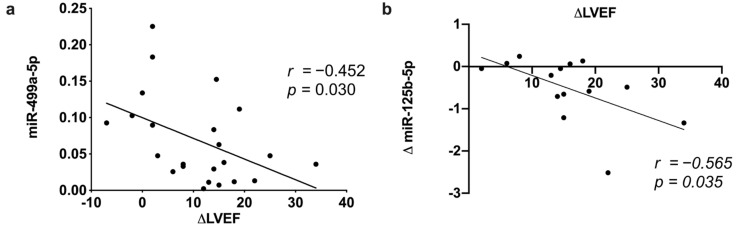
miRNA correlation with CRT response. Correlation between LVEF change and (**a**) baseline miR-499a levels and (**b**) follow-up miR-125b decrease.

**Figure 6 cells-11-00271-f006:**
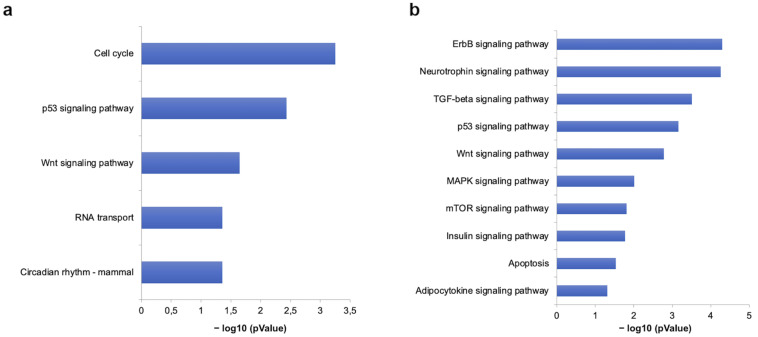
Gene set enrichment analysis of KEGG pathways of miR-499-5p and miR-125b-5p. Pathways modulated at baseline (**a**) and follow-up (**b**) by differentially expressed miRNAs, expressed as negative logarithm of *p*-value (−LogP).

**Figure 7 cells-11-00271-f007:**
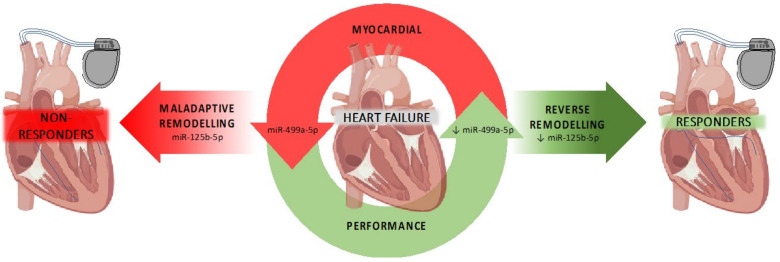
Graphical overview of hypothetical miRNA-regulated processes associated with CRT scheme.

**Table 1 cells-11-00271-t001:** Clinical parameters. Data are presented as mean + SD, % (*n*), or median and interquartile range (25–75%). * *p* < 0.05 baseline vs. 6 months and # *p* < 0.05 responders vs. non-responders.

	Total (*n* = 25)	Responders (*n* = 14)		Non-responders (*n* = 11)	
		Baseline	6 m	Baseline	6 m
Age, years		77 (±8)		75 (±5)	
Sex, male/female	15/10	9/5		6/5	
BMI, kg/m^2^	28.2 ± 0.83	28.3 ± 5.0	28.5 ± 5.6	28 ±3.18	28.8 ± 3.51
Systolic blood pressure	111 ± 15.62	115 ±19.42		111 ± 7.85	
Heart rate	73 ± 19.63	77 ±22		68 ±17	
Dyslipidaemia % (*n*)	44 (11)	29 (4)		64 (7)	
Diabetes, % (*n*)	36 (9)	43 (6)		27 (3)	
Hypertension, % (*n*)	68 (17)	71 (10)		64 (7)	
COPD (*n*)	25 (3)	7 (1)		18 (2)	
GFR mL/min	55 ± 22	56 ± 27	58 ± 21	53 ± 15	53 ± 24
NT-proBNP, pg/mL	4194 ± 4083	3361 ± 2697	1744 ± 1481 *	5254 ± 5325 #	12,914 ± 29,691 *
NYHA class, *n*	2.64 ± 0.64	2.43 ± 0.51	1.64 ± 0.50 *	2.91 ± 0.70	2.6 ± 0.97
Aetiology, % (*n*)					
Ischaemic		21.43 (3)		18.18 (2)	
Non-ischaemic		78.57 (11)		81.82 (9)	
Medical treatment, % (*n*)					
ACEIs/ARBs	56 (14)	50 (7)	36 (5)	64 (7)	45 (5)
Beta-blockers	84 (21)	86 (12)	93 (13)	82 (9)	91 (10)
ARNI	40 (10)	50 (7)	57 (8)	27 (3)	36 (4)
Spironolactone	80 (20)	79 (11)	79 (11)	82 (9)	82 (9)
Diuretics	88 (22)	79 (11)	64 (9)	100 (11)	64 (7)
Digoxin	20 (5)	21(3)		18 (2)	
Statin	56 (14)	50 (7)		64 (7)	
Echocardiographic data					
LVEDD (mm)	64 ± 10	65 ± 9	57 ± 7	63 ± 12	60 ± 9
LVESD (mm)	57 ± 12	59 ± 10	48 ± 8	55 ± 13	52 ± 10
LVEDVi, mL/m^2^	177 ± 87	187 ± 101	99 ± 44 *	164 ± 68	171 ± 74
LVESVi, mL/m^2^	129 ± 76	138 ± 87	59 ± 38 *	116 ± 61	119 ± 70
LVEF	31 ± 9	30 ± 8	46 ± 12 *	32 ± 10	35 ± 13

ACEIs, angiotensin-converting enzyme inhibitors; ARNI, angiotensin receptor-neprilysin inhibitors; BMI, body mass index; COPD, chronic obstructive pulmonary disease; GFR, glomerular filtration rate; LVEDD, left ventricular end-diastolic diameter; LVESD, left ventricular end-systolic diameter; LVEDVi, left ventricular end-diastolic volume index; LVESVi, left ventricular end-systolic volume index.

## Data Availability

The data presented in this study are available on Appendix A.

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
