# Peer review of "Circulating miR-499a and miR-125b as Potential Predictors of Left Ventricular Ejection Fraction Improvement after Cardiac Resynchronization Therapy"

_cells, 2022, doi:10.3390/cells11020271_

Round 1
Reviewer 1 Report
- Small number of patients in the study is the major issue for me. In addition to this, I believe that following issues need to be addressed:
- Define/explain Cardiac resynchronization therapy in introduction
- Rationale for choosing these 2 specific miRNAs should be provided in introduction
- Much more details in methodology should be provide (describe CST, when samples were exactly taken; at what times interventions were conducted; more details about RT-PCR (primers required etc.).
- On page 4 there is a large portion of text that looks like a part of instruction for authors; this should be deleted.
- In discussion, miRNA should be matched with non-miRNA HF biomarkers.
Author Response
We would like to thank the reviewer for their time spent on reviewing our manuscript and their meticulous comments, which help us to improve the article. All the suggestions and concerns from the referee have been addressed.
Response to Reviewer 1 Comments
- Small number of patients in the study is the major issue for me.
We fully agree with the reviewer, although our results are consistent and have reached statistical significance, the small number of patients has been discussed as a limitation at the end of discussion section lines 328-346:
“The present study has some limitations. First, the study was conducted in a single center with a small sample size. Since the study was performed to assess early biomarkers performance, follow-up period was extended until the moment in which patients could be classified by CRT response. Future studies should be extended up to 12 months to assess association between circulating miRNA changes and LVEF as well as long-term predictive capacity. Although all echocardiographic studies were always performed by the same experienced operator, accredited by the European Society of Cardiology in Echocardi-ography, and always with the same device, intraobserver error must always be considered in data interpretation. Also, although the patients were well characterized, clinical value of miRNAs as biomarkers for CRT success should be compared to classical pre-dictive parameters (sex, CVD, bundle branch block type, underlying rhythm etc.), which require a larger sample size. Our data are exploratory about the myocardial substrate characteristics that could be related to CRT response; however, for translation into clinical practice it would be necessary to propose a clinical trial, including a larger sample size and healthy controls, that could confirm our hypothesis. Also, our results suggest that selective miRNA regulation by using anti-miRNAs, miRNAs or miR-mimetic sponges might constitute a therapeutic approach. Whether these pathways truly constitute a therapeutic approach will need to be determined by laboratory experiments and is open for future studies.”
In addition to this, I believe that following issues need to be addressed:
We would like to thank the referee for her/his thorough review of the manuscript; we apologize for any lack of information or clarity noted by the reviewer. We certainly believe that her/his contribution has improved the understanding of the manuscript.
- Define/explain Cardiac resynchronization therapy in introduction
According to reviewer’s comment, we have included the definition as follows:
Lines 49-57: “When dyssynchrony is suspected in HF patients, cardiac resynchronization therapy (CRT) is emerging as a therapeutic approach to restore cardiac synchronization. This can be accomplished using biventricular pacing, this mean, a pacing lead in the right ventricle and a second lead in the lateral wall of the left ventricular (through an epicardial vein). By reestablishing cardiac synchrony, hemodynamics may be improved due to the optimization of the interventricular and intraventricular contraction sequences, thereby potentially decreasing mitral regurgitation, reducing pulmonary venous pressure, increasing myocardial preload, and improving cardiac output [2,3].”
- Rationale for choosing these 2 specific miRNAs should be provided in introduction
Although both microRNAs were chosen based on significant expression differences, subsequent analysis of ROC curves and its correlation with LVEF, we fully agree with the reviewer on the need to provide in the introduction a rationale as to why both miRNAs are relevant in an HF context. Relevance which is explained in-depth throughout the discussion in relation to our results.
Following the recommendations, we have introduced in the revised version the text as follows:
Lines 75-78: “Among dysregulated miRNAs in CVD, miR-499 and miR-125b-5p have been closely involved in HF. Both were described as highly expressed cardiac miRNAs, which are released under stressful conditions [10,21,22] as well as good circulating biomarkers in HF [22,23]”
- Much more details in methodology should be provide (describe CST, when samples were exactly taken; at what times interventions were conducted; more details about RT-PCR (primers required etc.).
Once again, we would like to thank the reviewer for the improvements that have been made as a result of the reviewer's accuracy, we have clarified the methodology section.
- Following the reviewer’s recommendation, CRT definition was previously introduced in the introduction section. Lines 49-57: “When dyssynchrony is suspected in HF patients, cardiac resynchronization therapy (CRT) is emerging as a therapeutic approach to restore cardiac synchronization. This can be accomplished using biventricular pacing, this mean, a pacing lead in the right ventricle and a second lead in the lateral wall of the left ventricular (through an epicardial vein). By reestablishing cardiac synchrony, hemodynamics may be improved due to the optimization of the interventricular and intraventricular contraction sequences, thereby potentially decreasing mitral regurgitation, reducing pulmonary venous pressure, increasing myocardial preload, and improving cardiac output [2,3].”
- Lines 136-138: “Blood samples were obtained in the morning of the CRT procedure, before device implantation, and 6 months after implantation using standard methods. Samples were processed and immediately frozen at -80º for 6 to 18 months until use”
- Lines 131-134: “Proper biventricular stimulation is systematically verified at discharge, at 3 and 6 months. All studies were performed in the morning, and medications were maintained according to current recommendation [29]”.
- Lines 140-154, RNA extraction and miRNA quantification were modified as follows:
“Total RNA was extracted using miRNeasy Serum/Plasma Advanced Kit (Qiagen, Hilden, Germany). Briefly, cDNA was transcribed from the extracted RNA using miScript II RT kit (Qiagen, Hilden, Germany) in a SimpliAmp Thermal Cycler (Applied Biosystems). The cDNA was pre-amplificated using miScript PreAMP PCR Kit (Qiagen, Hilden, Germany) with a miScript PreAmp Universal Primer and Human Cardiovascular Disease miScript PreAmp Pathway Primer Mix (MBHS-113Z, Qiagen, Hilden, Germany) and then, to perform miRNA quantification by polymerase chain reaction (RT-qPCR), was aliquoted into the Human Cardiovascular Disease miScript miRNA PCR Array containing 84 different predesigned mature miRNAs (MIHS-113Z, Qiagen, Hilden, Germany) with a miScript SYBR Green PCR kit (Qiagen, Hilden, Germany). miRNAs are listed in Supplemental Figure 1. All cDNA steps and PCR setup were performed by QuantStudio™ 7 Flex Real-Time PCR System, 384-well (Applied-Biosystems, CA, USA). PCR cycling was performed according to the manufacturer’s protocol and conditions. Individual miRNAs were determined to be detected when Ct values were lower than 30, Ct value ≥ 30 were considered not detected.”
- On page 4 there is a large portion of text that looks like a part of instruction for authors; this should be deleted.
Once again, we apologize for this mistake; we have deleted the paragraph.
- In discussion, miRNA should be matched with non-miRNA HF biomarkers.
We did not mention any association between dysregulated miRNAs and other classical markers of HF in the previous version of the manuscript because due to the lack of significant results. However, we fully agree with the reviewer that the lack of correlation must be also clarified, contributing to the interpretation of the study. Following reviewer´s recommendation, revised version of the manuscript have been modified as follows:
- Lines 302-303: “No other clinical biomarker correlates with miR-499-5p expression (baseline or follow-up).”
- Lines 305-306: “No significant associations were found between miR-125b-5p and other clinical biomarkers.”
Please see the attachment.

Reviewer 2 Report
The Authors aimed to study the predictive value of miRNA in patients receiving RCT. The Authors also performed a follow-up to study the changes of miRNA expression and to find a connection between miRNA and functional status of the patients.
The subject is well chosen and addresses a relevant issue. The manuscript reads well. Statistical methods are acceptable. Materials are overdetailed; however, this can be useful for future research. Results
Tables and Figures are illustrative and easy to understand.
However, some points need to be addressed in a revised manuscript before considering for publication.
Introduction
Line 72 Our data showed… this sentence is a bit confusing – in the Introduction section – I recommend to remove it or incorporate it within the Discussion.
Materials and methods
Line 83 citation
Line 101 Echocardiography. The manufacturer of the used echocardiography system is missing. What actions were taken to reduce interobserver error? (between baseline and 6m measurements).
Line 114 Blood collection
How much time elapsed between sampling and analysis?
How were the samples stored until analysis?
Line 140 “Paired and unpaired t-test….” How was the normality tested?
Results
Line 206 “…showed a marked sensitivity and specificity…” I recommend to express this numerically.
Line 223 “3.3. Pathways associated with differentially regulated miRNAs in CRT patients” Why is this section in the “Results” this is not an actual result of this study.
For clairvoyance and to help future research raw data and/or analysis should be included as supplementary.
Author Response
Since the response to the reviewer's comments include tables, please see the attached PDF.

Round 2
Reviewer 1 Report
Authors have amended manuscript as I requested.
This manuscript is a resubmission of an earlier submission. The following is a list of the peer review reports and author responses from that submission.